# Measurement of filtration efficiencies of healthcare and consumer materials using modified respirator fit tester setup

Kenneth D. Long[1,2]*, Elizabeth V. Woodburn[3], Ian C. Berg[1], Valerie Chen[3], William S. Scott[2,3,4]

1 Department of Bioengineering, University of Illinois at Urbana-Champaign, Urbana, Illinois, United States of America, 2 University of Illinois College of Medicine at Urbana, Urbana, Illinois, United States of America, 3 Carle Illinois College of Medicine, Champaign, Illinois, United States of America, 4 Carle Foundation Hospital, Urbana, Illinois, United States of America

* long6@illinois.edu

**Data Availability Statement:** All relevant data are within the manuscript and its Supporting Information files.

## Abstract

During the current SARS-CoV-2 pandemic there is unprecedented demand for personal protective equipment (PPE), especially N95 respirators and surgical masks. The ability of SARS-CoV-2 to be transmitted via respiratory droplets from asymptomatic individuals has necessitated increased usage of both N95 respirators in the healthcare setting and masks (both surgical and homemade) in public spaces. These precautions rely on two fundamental principles of transmission prevention: particle filtration and droplet containment. The former is the focus of NIOSH N95 testing guidelines, and the latter is an FDA guideline for respirators and surgical masks. While studies have investigated droplet containment to provide guidance for homemade mask production, limited work has been done to characterize the filtration efficiency (FE) of materials used in home mask making. In this work, we demonstrate the low-cost (<$300) conversion of standard equipment used to fit-test respirators in hospital and industrial settings into a setup that measures quantitative FEs of materials based on NIOSH N95 guidelines, and subsequently measure FEs of materials found in healthcare and consumer spaces. These materials demonstrate significant variability in filtration characteristics, even for visually similar materials. We demonstrate a FE of 96.49% and pressure drop of 25.4 mmH$_2$0 for a double-layer of sterilization wrap used in surgical suites and a FE of 90.37% for a combination of consumer-grade materials. The excellent filtration characteristics of the former demonstrate potential utility for emergent situations when N95 respirators are not available, while those of the latter demonstrate that a high FE can be achieved using publicly available materials.

## Introduction

The COVID-19 pandemic has created a surge in demand for personal protective equipment (PPE), most notably N95 respirators and surgical masks. While the virus is still being characterized, there exists uncertainty and conflicting recommendations from national and

**Funding:** The work described was supported by a Carle-Illinois College of Medicine Innovation Pathway Grant (medicine.illinois.edu). IB was supported by the National Institute of Biomedical Imaging and Bioengineering of the National Institutes of Health (www.nibib.nih.gov/) under Award Number T32EB019944. The content is solely the responsibility of the authors and does not necessarily represent the official views of the National Institutes of Health. The funders had no role in study design, data collection and analysis, decision to publish, or preparation of the manuscript.

**Competing interests:** The authors have declared that no competing interests exist.

international agencies regarding airborne and droplet transmission and the appropriate PPE for different levels of contact in the healthcare setting [1–4]. Furthermore, there is ongoing academic discourse about the definition of terms such as 'droplet', 'droplet nuclei', and 'aerosol' for purposes of respiratory and aerosol transmission [5–7]. While much of this is beyond the scope of this article, it is important to recognize the distinction between respirators (such as the N95) and masks. While a respirator is designed to protect the wearer from inhalation of particles and aerosols that may be infectious or otherwise harmful, a mask is used for situations where droplet protection is sufficient [8]. Current CDC guidance calls for the use of respirators and medical facemasks in healthcare settings with respiratory protection programs and the use of face coverings for all individuals in public settings [1].

Increased demand for respirators has put significant stress on the supply chain with many reports of healthcare professionals (HCPs) forced to work without appropriate PPE [9–11]. Rapid responses by regulatory bodies have tried to keep up with demand by broadening categories of respirators approved for healthcare settings, providing emergency use authorizations of sterilization techniques, and creating guidelines for reuse of previously disposable faceplate respirators [12–14]. Contingency recommendations for HCP prioritization are laid out in CDC guidance [1]. We propose that, during instances of N95 shortage, it may be possible to construct a respirator that is more effective than a surgical mask using materials available in healthcare settings. The main barrier to designing such a respirator, however, is the lack of data on material properties that would make them suitable for this use.

Similarly, when the CDC recommended face coverings for public use in March 2020, Americans began a nationwide effort to produce cloth masks at home to preserve supplies of medical-grade PPE for healthcare workers, largely without data to inform their design or material selection [15]. The national, state-level, and local guidance for public mask donning is primarily to prevent the spread of the wearer's own droplets (containment strategy), and typically endorse a range of do-it-yourself (DIY) mask construction techniques and materials [8]. From a public health standpoint, prevention of disease transmission is the primary goal, similar to that in the healthcare setting.

Three components determine the effectiveness of a face covering for transmission prevention: (1) the "fit factor," which measures how well the mask seals to the face, (2) the filtration efficiency (FE), a material's intrinsic ability to filter out particles in the air, and (3) the fluid resistance, a material's ability to prevent penetration of fluids at pressures found in the human body. Medical respirators must meet metrics for all three, while surgical masks need only meet the latter two, with a lower threshold for filtration efficiency [16]. Though no standards currently exist for DIY masks, the same metrics are appropriate for comparative effectiveness. Other work has made inroads on mask fit and fluid resistance, but there is still substantial need for quantitative FE data to inform DIY mask construction [17–19]. The recent resurgence of universal mask usage globally has renewed interest in quantitative measurements and comparisons of materials for their design [20, 21].

In the United States, the National Institute for Occupational Safety and Health (NIOSH) is the premier agency that verifies filtration capabilities of respirators using specially designed equipment that can assess fit and filtration [8, 22]. Unfortunately, labs with such specialized equipment are in high demand, with wait times of up to a month [23]. The increase in demand for these measurements is a direct reflection of their importance for quantitative benchmarking of filtration efficiency to assess strategies for identifying new sources, reusing existing models, and different sterilization techniques to extend existing respirator supplies. Much of the work to date has focused on sterilization of respirators for subsequent reuse. While a few studies have been completed and submitted to preprint servers after waiting for data from extant labs [24–29], a number of other preprints [30, 31] and publications [32–34] have developed

apparatuses to measure filtration efficiency as best they can independently. There is significant heterogeneity in these setups; some are built with assistance from labs which study aerosols, others with equipment found on hand. These studies typically look for a decrease in measured filtration efficiency (or surrogate endpoint) and use non-inferiority as an endpoint.

The identification of alternative materials and/or combinations thereof is a more challenging proposition and requires a more rigorous setup as there is not a built-in baseline for relative comparison. As might be expected, less work has been published thus far, likely a reflection of this challenge. Frequently cited studies looking at common household textiles from 2013 and 2010 demonstrate the variation present in sometimes similar-appearing fabrics [35, 36]. When paired with recent studies looking at differences in droplet penetration and fluid resistance [17, 37, 38], suggesting more accessible techniques may provide further opportunities for material analysis. During the current pandemic, though, *de novo* development of filtration efficiency measurement setups has been limited largely to national labs [39, 40].

With so many potential avenues for research into alternative materials, sterilization, reuse, and even fabrication for respirators, let alone the design and filtration properties of masks, there is significant need for an easily-accessible setup for filtration efficiency measurement as an enabling tool for further research on many of these fronts. Here we describe a setup that uses a standard quantitative respirator fit test machine augmented with low-cost hardware (<\$300) to measure the FE of materials using methods modeled after NIOSH guidelines. We obtain the FE and pressure drop ($\Delta$p) of 11 hospital-grade and 10 consumer-grade materials, and, based on their performance, identify the most promising materials for homemade masks when commercial N95 respirators are unavailable.

## Methods

### Testing setup

The test setup was developed based on NIOSH testing procedure TEB-APR-STP-0059, which is used to evaluate the FE of N95 respirators [22, 41]. In this procedure, air containing aerosolized NaCl particles is pumped through the test sample material at 85 Lpm. Particle concentration is measured upstream and downstream of the filter using photometers and the FE is calculated as the percentage of particles blocked by the respirator. This test is done using an Automated Filter Tester (TSI 8130 Shoreview, MN) [41].

Our test setup replicates this measurement using a particle generator and respirator fit tester, which are often available in hospitals and occupational safety organizations that perform quantitative fit testing (Fig 1) [42]. A particle generator (TSI 8026 Shoreview, MN) generates aerosolized NaCl particles with a median diameter of 0.04 microns and geometric standard deviation (GSD) of 2.2 in a chamber (plastic storage bin) to provide a steady-state supply of particles at sufficient concentration (>4000 particles/cm$^3$) [43]. While the NIOSH test uses NaCl aerosol with a median diameter of 0.075 micron and GSD of less than 1.86, the smaller particles generated in this setup provide a suitable measurement of FE since they are within the typical maximally penetrant range of particle size for N95 respirators (.03-.06 microns) [36, 41].

An air-tight material sample mount is formed using two stainless steel 3" to 1.5" bowl reducers, a gasket with tri-clamp sanitary fittings, and 1.5" tri-clamp to 3/8" hose barbed adapters at each end for connection to rubber hose. Airflow is provided by the hospital's house vacuum supply. Flow is controlled and measured by an inline flowmeter. Testing was performed at 25 Lpm unless otherwise noted. The sample area exposed to flow is 41.7 cm$^2$. NIOSH testing is performed on entire respirators, which have variable surface areas that are typically around 150 cm$^2$ [16]. At 25 Lpm, these samples experience linear flow velocities equal to or greater

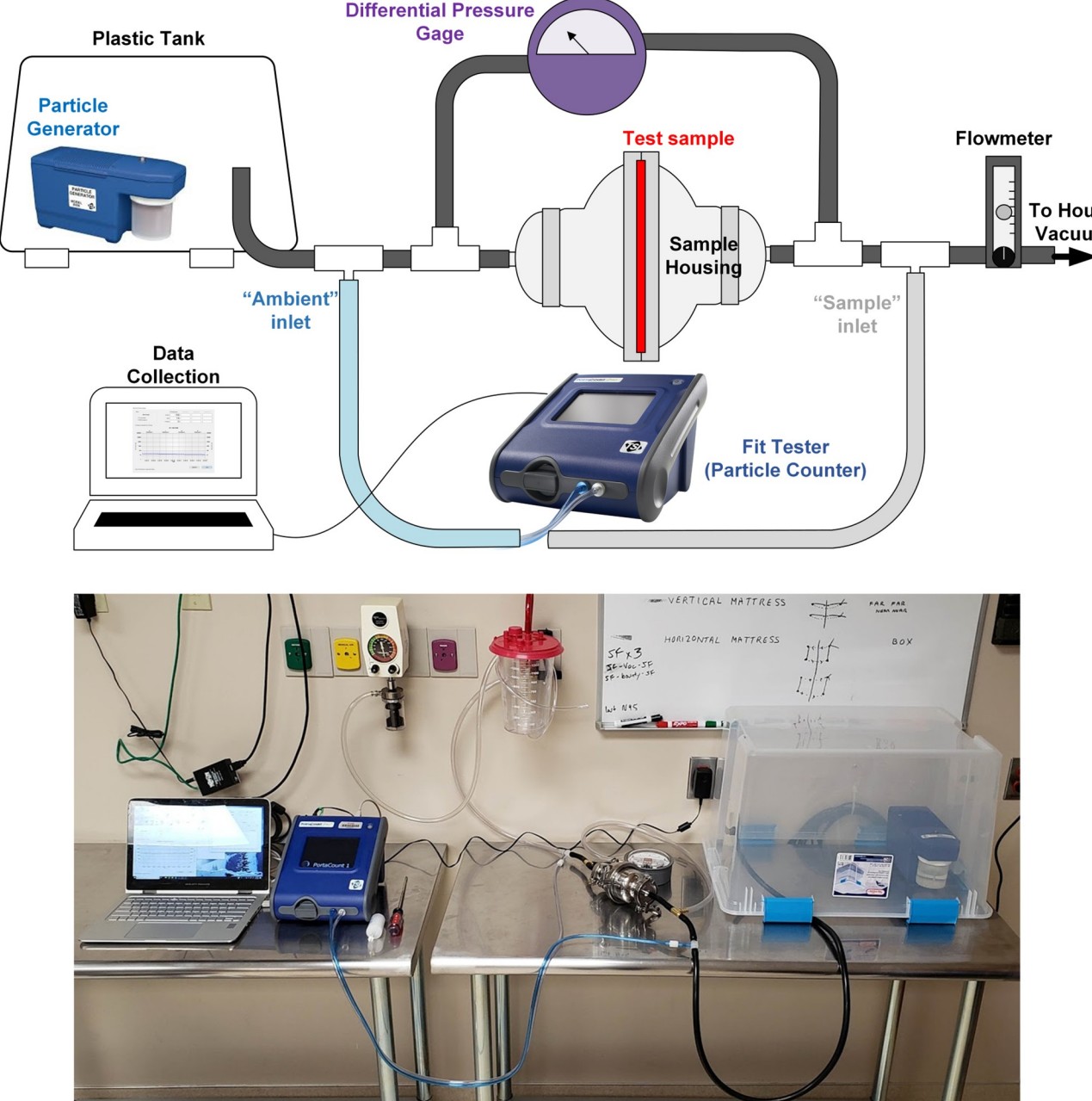

**Fig 1. Filtration efficiency measurement setup.** Diagram of filtration efficiency measurement setup, design utilizing hospital fit testing apparatus, particle generator, and house vacuum supply augmented with low-cost hardware.

than that experienced by a full respirator at 85 Lpm, making this an appropriate flow rate to approximate the NIOSH methodology given our smaller sample surface area.

Particle concentration is measured using a PortaCount Respirator Fit Tester (TSI 8038, Shoreview, MN) capable of detecting particles between 0.02 and 1 micron. When used for fit-factor testing, the sample inlet is attached to a respirator to sample air the user breathes inside the respirator, while the ambient inlet samples the air outside the respirator. The instrument alternates measuring particle concentrations from the sample and ambient lines at a rate of 1.0

Lpm and calculates the fit factor as the ratio of the two. Our setup uses T fittings to connect the ambient inlet immediately upstream of the sample housing to the sample inlet immediately downstream. One-way flow valves at each inlet protect the instrument from possible suction from the house vacuum. The PortaCount is connected to a computer running FitPro + software, which enables manual control of the active inlet and real-time monitoring of particle concentration. Since the software does not provide continuous particle concentration data logging, we developed a MATLAB script to record the "Real Time Display" module and record each reading using optical character recognition. For FE testing, upstream concentration is measured for one minute, followed by downstream concentration measurements for two minutes and upstream concentration measurement for one additional minute. During the switchover from upstream to downstream measurement, there is a break in data collection followed by a ramp up period as sampling restarts. The datapoints corresponding to the break and ramp up periods are manually cropped out during analysis. The upstream and down particle concentrations are averaged over the collection time, and the FE is calculated using

$$\text{FE} = \left(1 - \frac{\text{average downstream concentration}}{\text{average upstream concentration}}\right) \times 100.$$

Replicate measurements are performed on separate material samples. FE measurements are reported as the mean FE of replicate measurements with standard deviations.

All components upstream of particle sampling are connected by anti-static hosing to minimize particle loss due to static adhesion. To characterize the particle source behavior and particle loss between sampling points, we run triplicate tests without filtration. There was no statistically significant difference between the measured particle concentration upstream or downstream of the filter housing. Average particle loss was 1.75% ± 2.32.

Air flow resistance testing is an additional requirement of respirator approval to demonstrate the respirator does not overly inhibit breathing. To do this, we install a pressure differential gauge across the sample housing. The pressure difference (Δp) at the testing flow rate is first measured without sample material, which is subtracted from material sample pressure differentiation measurements to account for Δp caused by the hardware. All Δps were measured at 25 Lpm unless otherwise noted. A list of parts and equipment used in the setup is provided in S1 Table.

## Sample material preparation

This study tests sample materials available in the hospital (healthcare-grade) and those accessible to the public (consumer-grade). The consumer-grade materials were selected due to reported usage in online mask-making tutorials or similarity to component textiles from commercial N95 respirators, such as nonwoven polypropylene (NWPP) [8]. Hospital grade materials were chosen by surveying medical catalogs and the hospital inventory. Materials were chosen based on their resemblance to the materials used in respirators or surgical masks. We also aimed to identify materials used for filtration, as personal protection equipment, or as splash or fluid barriers for equipment.

Samples are cut to 91 mm circles to fit in the housing. Materials with a filtration efficiency greater than 40% are tested twice more using two additional samples. Since masks are rarely made of a single layer of material, combinations of materials are also tested. Combinations were similarly selected based on found online mask-making tutorials [44–46]. A complete list of tested materials is provided in S2 Table.

## Statistical methods

Tests for significance were performed using a two sample t-test with p value of 0.05 using MATAB software. FE data are presented as the average of replicate FE measurements, with standard deviation included for materials with at least three replicates. Whiskers on boxplots indicate maximum and minimum FE measurements for each material.

## Results

Samples prepared from 3M 1870 and 1860 N95 respirators demonstrated FE of 99.43% ± 0.18 and 98.89% respectively (Fig 2), providing validation that the setup can identify N95-certified materials. A National Personal Protective Technology Laboratory (NPPTL) study reported filtration efficiencies of 99.63% and 98.47% respectively for these two N95 models [47]. Furthermore, a recalled N95 respirator (HY8510, Tronex) had a measured FE of 92.85%. An NPPTL study reported a filtration efficiency between 82.8 and 94% for this model across 20 measurements, with the mean filtration efficiency being 91.09% [48]. The consistency of our measurements to those reported elsewhere serve to validate the accuracy of this set up. Further, the difference in measurements of approved and recalled N95s demonstrates an ability to discern between even high-performance materials.

The FE for consumer grade materials ranged from 35 and 53%. Two materials (coffee filter and cotton cloth) measured notably lower while the vacuum bag performed much better with

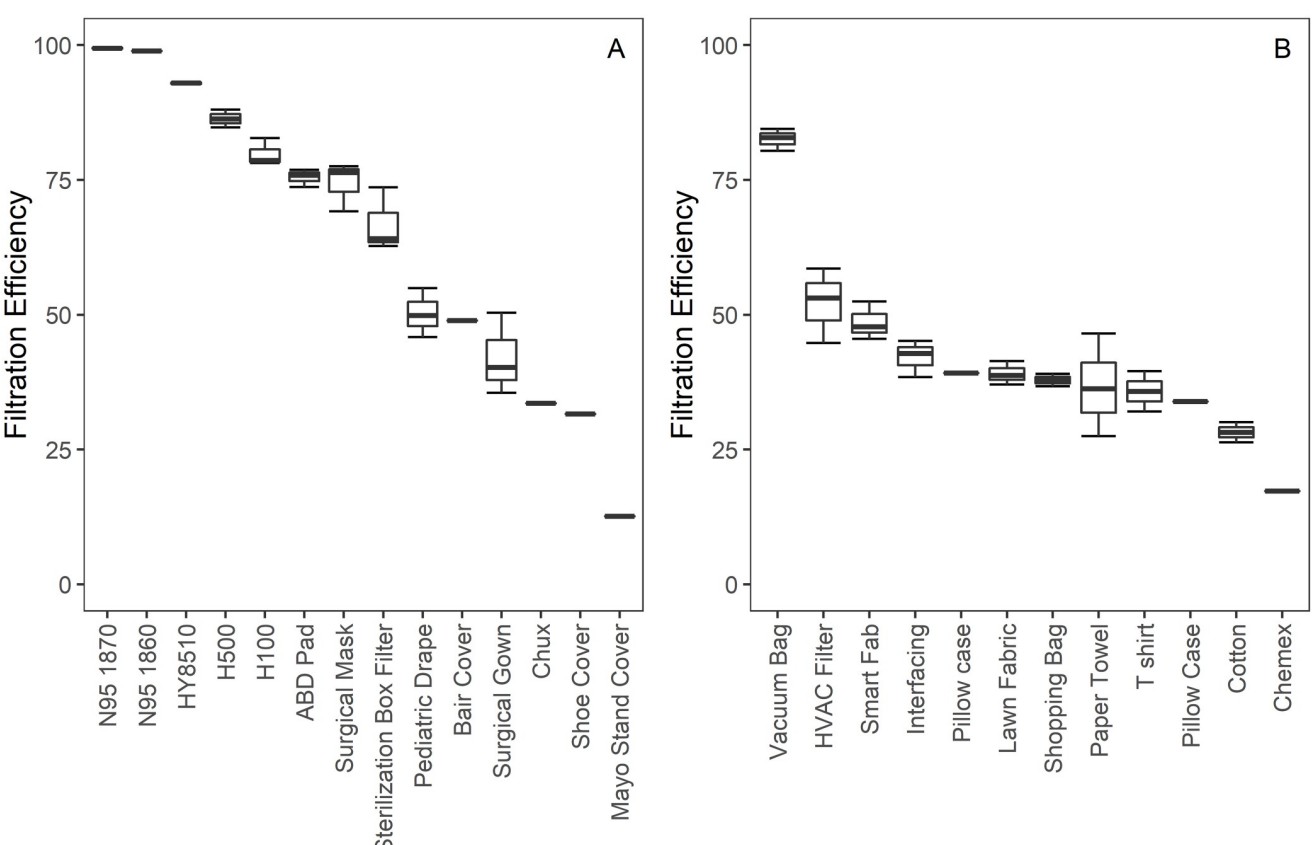

**Fig 2. Filtration efficiency results.** Filtration efficiencies for (A) hospital-grade materials and (B) consumer-grade materials, with quartile data reported for materials where multiple samples were tested.

**Table 1. Filtration efficiencies and pressure drops of material combinations.**

| Materials | FE (%) | $\Delta$p (mmH$_2$O) |
|---|---|---|
| H500/H500 | 96.49 | 25.40 |
| H500/H100 | 93.87 | 12.70 |
| Smart-Fab/Vacuum Bag/Smart-Fab | 90.37 | 10.16 |
| Cotton/Vacuum Bag/Cotton | 83.38 | 15.24 |
| 3x Smart-Fab | 79.92 | 7.62 |
| Smart-Fab/Paper Towel /Smart-Fab | 73.53 | 10.16 |
| 3x Shopping Bag | 59.75 | 6.35 |
| Cotton/Paper Towel/Cotton | 34.76 | 10.16 |
| 2x Cotton | 14.90 | 6.35 |

a FE of 82% ±2.06. The FE of medical grade materials were more linearly distributed between 30 to 86%. The most effective medical-grade material was heavyweight sterilization wrap (H500, Halyard) with a FE of 87%±1.76).

No single material tested measured at the N95 FE level of 95%. Multilayered combinations of materials were also tested (Table 1). A sample consisting of three layers of NWPP art fabric (Smart-Fab Inc.) achieved a FE of 80%. While increasing the number of layers may produce a greater FE, it also increases manufacturing difficulty and decreases breathability. Notably a FE of 96%, which exceeds the threshold for an N95 certification, was achieved with two layers of heavyweight sterilization wrap. While the lightweight sterilization wrap (H100, Halyard) FE differed from that of the heavyweight wrap by 7.8%, a combination of lightweight and heavyweight achieved a FE of nearly 94%. These results suggest that the FE of material combinations cannot be predicted by simple additive or multiplicative relationships.

Filtration efficiencies (FE) and pressure drops ($\Delta$p) of combination samples of two to three layers of material.

Pressure drop results are provided in Table 2. At 25 Lpm, the maximum measured $\Delta$p for the N95 respirator samples was 10.16 mmH$_2$O. Most materials had a maximum measured $\Delta$p equal to or less than this value. The remaining materials had measured maximum $\Delta$p values of 12.7 mmH$_2$O. Combination material $\Delta$p results are provided in Table 2. All measured $\Delta$p values were below the maximum NIOSH-allowed resistance for respirators, 35 mmH$_2$O (42 CFR 84.180). These data suggest than all tested materials have a suitably low air flow resistance. Because NIOSH testing is completed at a challenge flowrate of 85 Lpm, additional air flow resistance testing on respirator candidate materials in their assembled respirator form may be necessary, though pressure drops less than that of the N95 respirators suggest that many materials would pass such testing.

## Discussion

In this study, we develop a setup that approximates the FE and airflow resistance testing requirements established by NIOSH, providing an approximate measurement of filtration properties when industry-grade testing is not available. It creates a steady quantity and distribution of particles, with negligible loss between the ambient and filtered sampling ports and provides flow velocities equivalent to those used in NIOSH testing, with simultaneous measurement of $\Delta$p and FE.

The measured filtration efficiencies of two N95 respirators were found to match those reported by a recent NIOSH publication looking at performance of stockpiled respirators [47]. The FE of the 3M 1870 was measured at 99.43% ± .18, and that of the 3M 1860 was measured

**Table 2. Measured pressure drops for single layers of tested materials.**

| Material | Flowrate (Lpm) | $\Delta p$ (mmH$_2$O) |
|---|---|---|
| **N95** | 10 | 2.54 |
| | 25 | 10.16 |
| | 40 | 6.35 |
| | 60 | 22.86 |
| **H500** | 25 | 22.86 |
| **Vacuum Bag** | | 12.7 |
| **H100** | | 10.16 |
| **Abdominal Pad** | | 10.16 |
| **Surgical Mask** | | 6.35 |
| **Sterilization Box Filter** | | 12.7 |
| **HVAC Filter** | | 0 |
| **Pediatric Drape** | | 12.7 |
| **Bair Cover** | | 10.16 |
| **Smart Fab** | | 12.7 |
| **Interfacing** | | 2.54 |
| **Lawn Fabric** | | 10.16 |
| **Shopping Bag** | | 10.16 |
| **Paper Towel** | | 7.62 |
| **Pillowcase** | | 12.7 |
| **T shirt** | | 12.7 |
| **Chux** | | 7.62 |
| **Shoe Cover** | | 6.35 |
| **Cotton** | | 6.35 |
| **Chemex** | | 12.7 |
| **Mayo Stand Cover** | | 10.16 |

Measured pressure drops ($\Delta p$) for single layers of tested materials at the listed flowrate in liter per minute (Lpm).

at 98.89%, which correlate well with NIOSH measured 99.63% and 98.75%, respectively (statistical comparison not possible with graphical NIOSH data) [47]. These data suggest that our setup provides a robust replication of NIOSH test standards. Furthermore, a recalled N95 from Tronex was measured at a FE of 92.95%, which can be compared with NIOSH measured FEs of 71–85% [48]. This comparison is challenging since these masks do not have printed manufacture dates or shelf lives and NIOSH results were from a single lot with unknown storage duration and conditions [48]. Additionally, our test sample intentionally minimized the presence of seams present in our smaller cross-sectional sample area, possibly contributing to a better observed performance. Similarly, the measured filtration efficiency of surgical masks, 74.36 ±4.54% correspond to values reported elsewhere [16]. With individual institutions sourcing respirators from a myriad of domestic, international, donated and stockpiled sources, the ability to perform in-house FE testing of received materials could create increased confidence in the performance of materials without the expense or delay of official confirmatory testing from a national lab particularly against a backdrop of some emergently-approved respirators being recalled by the FDA after in-house testing [49].

The samples tested represent a variety of both healthcare-grade and consumer-grade materials. While not exhaustive, they demonstrate a range of filtration efficiencies and show that it is difficult to predict filtration performance using visual inspection. Several NWPP materials (Smart-Fab, lawn fabric, reusable shopping bag, interfacing) appear nearly identical but have

FEs from 38% to 49%. Conversely, the abdominal pad and sterilization box filter perform similarly despite their dramatic difference in thickness.

During an emergent shortage of N95 respirators sterilization wrap offers a promising alternative for makeshift or handmade N95 respirators in healthcare settings. Two layers of sterilization wrap achieve filtration of 96.49% of test particles, demonstrating potential for use in constructing an N95 alternative. As most respirators and surgical masks are constructed from at least 3 layers of material, this two-layer combination is a promising possibility for further investigation, even if an additional barrier layer is needed for droplet protection. Other work tests materials for droplet permeability, and several individuals and institutions are developing sewing or heat-sealing patterns for mask fabrication that can achieve an air-tight fit [17]. This study provides needed insight on appropriate material selection for use in such designs.

Mask wearing in public spaces has evolved from a recommendation to requirement in communities across the country, leading to a myriad of homemade masks constructed largely without consideration for FE or droplet containment [50]. While recent studies have provided data on droplet protection in homemade masks [17] FE measurements may be cost-prohibitive and require equipment that is in short supply. Studies show that the main benefit of masks is not in protecting the wearer but from preventing the wearer from transmitting infectious respiratory material [35]. This is because of limitations to current masks, which are rarely airtight and often designed for convenience and comfort rather than their proven ability to stop disease transmission. If mask materials are selected for FE and droplet containment properties, they may provide benefit to the wearer beyond source control even if they do not achieve FEs as high as 95% [51].

## Limitations

A known limitation of the presented setup when compared to the NISOH specifications is the smaller cross-sectional area of filter tested. This was a practical limitation of our experimental work, as we wanted to measure samples of N95 respirators used in our hospital. The minimum dimension of the 1070+ respirator is approximately 76 mm, so we selected this size of sanitary fitting for our sample mount. Related to this deviation from NIOSH guidelines, we completed all FE testing at a flow rate of 25 Lpm to keep the linear face velocity the same as the NIOSH testing setup, as our sample cross sectional area was approximately 28% that of commercial respirators [16].

Another limitation of our setup is the lack of charge neutralization. NIOSH regulations specify that NaCl aerosols should be neutralized prior to filtration. Neutralization is typically performed with equipment containing radioactive isotopes, such as Polonium 210, and cost hundreds of dollars. While particle neutralization can decrease overall filter efficiency in a material-dependent manner, these effects are most significant at larger particle sizes and higher charges. Studies show minimal decrease in FE with uncharged particles when compared to charged particles [52, 53]. Taken together, this suggests that our study may overestimate FE, though the error is likely small given the consistency of our measurements with those reported elsewhere for surgical masks and respirators.

Lastly, this study focuses on the FEs of mask materials without addressing fit. Our results show that certain material combinations can achieve >95% FE, but studies have shown that even a minimal compromise in the seal can cause a significant loss in filtration capability [54–56]. Further studies are needed to assess which home-made mask designs can achieve an appropriate seal.

## Conclusions

In this work we demonstrate that alternative mask materials can be used in combination to achieve filtration efficiencies that approach those of N95 respirators. When rapid in-house testing of material filtration is needed, hospitals with quantitative fit-testing equipment can create a setup that approximates NIOSH testing standards using <$300 of additional equipment. The setup presented demonstrated filtration efficiency measurements of respirators nearly identical to those measured by NIOSH during the current pandemic and produced measurements of other materials similar to those reported elsewhere in the literature. Measurement of healthcare and consumer materials demonstrates that there is wide variation in filtration properties that may not be apparent on visual inspection. During the current SARS-2-CoV pandemic, the rapid increase in PPE usage demonstrates the importance of understanding intrinsic material properties to inform emergent fabrication of face coverings in both healthcare and consumer settings. Hospitals and other occupational safety organizations may already have the equipment on-hand to provide quantitative measurements of material filtration efficiencies as a beneficial service to their community in times of need.

## Supporting information

**S1 Table. Parts and equipment used in test setup.** Full list of and details on equipment and parts used in measurement set up.
(DOCX)

**S2 Table. Material details and results.** Complete list of and details of materials tested, and filtration efficiency and pressure drop results.
(DOCX)

**S1 File. Complete filtration efficiency results.** Individual filtration efficiency measurement data. Mean and standard deviation upstream and downstream data describe the particle concentration data measured across the time spans described in methods.
(XLSX)

## Acknowledgments

The authors wish to express appreciation to Joseph V. Puthussery for expertise related to test setup design, and to Dr. Blair Rowitz for assistance in facilitating testing facility access and institutional support at Carle Foundation Hospital.

## Author Contributions

**Conceptualization:** Elizabeth V. Woodburn, Ian C. Berg.

**Data curation:** Ian C. Berg.

**Formal analysis:** Kenneth D. Long, Elizabeth V. Woodburn, Ian C. Berg, Valerie Chen.

**Funding acquisition:** Elizabeth V. Woodburn, Valerie Chen.

**Investigation:** Kenneth D. Long, Elizabeth V. Woodburn, Ian C. Berg, Valerie Chen.

**Methodology:** Kenneth D. Long, Elizabeth V. Woodburn, Ian C. Berg, Valerie Chen.

**Project administration:** Kenneth D. Long, Elizabeth V. Woodburn, Valerie Chen.

**Resources:** Kenneth D. Long, Elizabeth V. Woodburn, Valerie Chen.

**Software:** Ian C. Berg.

**Supervision:** William S. Scott.

**Validation:** Kenneth D. Long, Elizabeth V. Woodburn, Ian C. Berg.

**Visualization:** Ian C. Berg.

**Writing – original draft:** Kenneth D. Long, Elizabeth V. Woodburn, Ian C. Berg, Valerie Chen.

**Writing – review & editing:** Kenneth D. Long, Elizabeth V. Woodburn, Ian C. Berg, Valerie Chen, William S. Scott.

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
