## [Decision Letter · Decision Letter 0]

26 Aug 2020

PONE-D-20-22953

Measurement of filtration efficiencies of healthcare and consumer materials using modified respirator fit tester setup

PLOS ONE

Dear Dr. Long,

Thank you for submitting your manuscript to PLOS ONE. After careful consideration, we feel that it has merit but does not fully meet PLOS ONE’s publication criteria as it currently stands. Therefore, we invite you to submit a revised version of the manuscript that addresses the points raised during the review process.

We look forward to receiving your revised manuscript.

Kind regards,

Ming Zhang, Ph.D.

Academic Editor

PLOS ONE

Journal Requirements:

2. Please check the filtration efficiency of the recalled Tronex N95 respirator. In the results section it is given as 94%, but in the discussion section and in Table S2 it is given as 92.95%.

Additional Editor Comments:

According to the reviewers' advice, I recommend to accept this paper after major revision.

Reviewers' comments:

Reviewer's Responses to Questions

**Comments to the Author**

1. Is the manuscript technically sound, and do the data support the conclusions?

Reviewer #1: Yes

Reviewer #2: Yes

2. Has the statistical analysis been performed appropriately and rigorously? 

Reviewer #1: Yes

Reviewer #2: Yes

3. Have the authors made all data underlying the findings in their manuscript fully available?

Reviewer #1: Yes

Reviewer #2: Yes

4. Is the manuscript presented in an intelligible fashion and written in standard English?

Reviewer #1: Yes

Reviewer #2: Yes

5. Review Comments to the Author

Reviewer #1: The paper describes a setup that uses a quantitative respirator fit test machine augmented with low-cost hardware to measure the FE of materials using methods modeled after NIOSH guidelines, and identify the most promising materials for homemade. There are still shortcomings in this research, which needs to be revised.

(1) In the Introduction, the paper lacks an overview of the existing literature. In the introduction, the authors only introduce the functions and shortages of masks. I suggest that the author review the existing literature. Although COVID-19 is a sudden pandemic, the author should review the existing literature from the perspective of the pandemic. In addition, the authors should supplement the main content and innovation of the research in the introduction.

(2) In the Methods part, the authors emphasize the Matlab or R software. The authors do not list any formulas for model analysis, only the software. The software is only used to analyze the model. It does not matter what software the author uses to complete this analysis. In the Statistical Methods part, the author should list the specific formulas for analysis. So, it is recommended that the authors should list relevant formulas in the Methods part.

(3) In the Sample Material Preparation part, the authors list all the materials in Table S2. Why do the authors choose these materials? What are the differences and connections between these materials? It is recommended that the authors should give a detailed explanation in the Sample Material Preparation part.

(4) In the Results part, the paper focuses on measuring the filtration efficiency of masks. The filtering efficiency of the mask is known, which is issued by an authoritative company. The filter efficiency value issued by the mask company is more convincing. How to make people more convinced of your results?

(5) In the Conclusions part, the authors said that “In this work we demonstrate that alternative mask materials can be used in combination to achieve filtration efficiencies that approach those of N95 respirators”. The practical significance of research conclusions needs to be more clear. In reality, few people wear several layers of mask materials to achieve the effect of N95. So, the practical significance of this conclusion needs to be supplemented.

(6) In the Conclusions part, the authors said that “When rapid in-house testing of material filtration is needed, hospitals with quantitative fit-testing equipment can create a setup that approximates NIOSH testing standards using <$300 of additional equipment”. Whether the test result is accurate? The paper should provide some evidence. Some analysis should be added in the Results part.

Reviewer #2: The manuscript focused on the measurement of filtration efficiencies using modified respirator fit tester setup. The research is technically sound and properly organized. The experiments are conducted rigorously and the conclusions are drawn appropriately. Add some keywords if necessary.

6. PLOS authors have the option to publish the peer review history of their article (what does this mean?). If published, this will include your full peer review and any attached files.

Reviewer #1: No

Reviewer #2: No

---

## [Author Response · Author response to Decision Letter 0]

12 Sep 2020

Editor’s comment: Please check the filtration efficiency of the recalled Tronex N95 respirator. In the results section it is given as 94%, but in the discussion section and in Table S2 it is given as 92.95%.

We appreciate the editor’s keen eye, and have corrected the body of the manuscript in the results section to reflect the appropriate number. 

Reviewer #1: The paper describes a setup that uses a quantitative respirator fit test machine augmented with low-cost hardware to measure the FE of materials using methods modeled after NIOSH guidelines, and identify the most promising materials for homemade. There are still shortcomings in this research, which needs to be revised.

(1) In the Introduction, the paper lacks an overview of the existing literature. In the introduction, the authors only introduce the functions and shortages of masks. I suggest that the author review the existing literature. Although COVID-19 is a sudden pandemic, the author should review the existing literature from the perspective of the pandemic. In addition, the authors should supplement the main content and innovation of the research in the introduction.

The authors appreciate the reviewer’s recommendations and have provided additional paragraphs surveying the contemporary landscape, with a particular focus on work related to filtration efficiency measurement, which there has been limited research thus far. The authors appreciate the opportunity to provide additional references to recent work in the area. For full transparency with regard to the state of peer review for cited material, the authors have been careful to indicate where recent preprints were cited, but preprints were included, as the author does agree with the reviewer that the landscape is changing rapidly.

(2) In the Methods part, the authors emphasize the Matlab or R software. The authors do not list any formulas for model analysis, only the software. The software is only used to analyze the model. It does not matter what software the author uses to complete this analysis. In the Statistical Methods part, the author should list the specific formulas for analysis. So, it is recommended that the authors should list relevant formulas in the Methods part.

The authors appreciate the reviewer’s recommendation to include more detail on the methods used in analysis. In the results section we have added additional detain on how data was collected, processed, and averaged over the collection time. We also added the formula for filtration efficiency. The authors appreciate drawing our attention to clarify the statistical analysis. We updated the text to now accurately describe that the whiskers identify the maximum and minimum FE measurements. We clarified that the test for significance was completed using a two sample t test function with p value of 0.05. We agree that the software used in this case does not matter but we included this information for clarity. We did not include the t test formula as it is a standard formula available in the software documentation and elsewhere. 

(3) In the Sample Material Preparation part, the authors list all the materials in Table S2. Why do the authors choose these materials? What are the differences and connections between these materials? It is recommended that the authors should give a detailed explanation in the Sample Material Preparation part.

The authors appreciate the recommendation that more detail be provided about what informed the material choices. We included a sentence regarding how the consumer grade materials were chosen. We added to this paragraph more detail on how we surveyed available hospital grade materials, and what criteria we looked for in identifying materials for testing

(4) In the Results part, the paper focuses on measuring the filtration efficiency of masks. The filtering efficiency of the mask is known, which is issued by an authoritative company. The filter efficiency value issued by the mask company is more convincing. How to make people more convinced of your results?

While the authors acknowledge that companies provide some (albeit minimal) information about filtration efficiencies of their respirators (rarely do they provide more than certification that their respirators meet N95 standards), the authors would pose that the appropriate authority for such determinations is NIOSH. As part of the NPPTL Respirator Assessments to Support the COVID-19 Response (https://www.cdc.gov/niosh/npptl/respirators/testing/NonNIOSHresults.html), NIOSH has produced a significant number of studies measuring existing respirators from various manufacturers, which the authors would suggest is the best comparator for our measurements. In fact, the data collected on our in-house system demonstrates a high-degree of agreement with those from NIOSH, including similar filtration efficiencies for 3M 1860 and 1870 respirators, and similarly-measured substandard performance for Tronex HY8510. 

In the discussion section we note, “the FE of the 3M 1870 was measured at 99.43% ± .18, and that of the 3M 1860 was measured at 98.89%, which correlate well with NIOSH measured 99.63% and 98.75%, respectively (statistical comparison not possible with graphical NIOSH data)”. We believe that the high degree of correlation between these measured filtration efficiencies, even at the upper limit of filtration capabilities, demonstrates a robust concordance between the two testing methodologies, as that range is where differences would be most pronounced. We have added the NPPTL citation in the Results section as well, to add clarity.

(5) In the Conclusions part, the authors said that “In this work we demonstrate that alternative mask materials can be used in combination to achieve filtration efficiencies that approach those of N95 respirators”. The practical significance of research conclusions needs to be more clear. In reality, few people wear several layers of mask materials to achieve the effect of N95. So, the practical significance of this conclusion needs to be supplemented.

The authors point to the construction of standard commercial masks and surgical masks, which are typically comprised of 2 and 3 layers, respectively. Similarly, a number of semi-professional handmade masks consist of multiple layers, either for the purposes of aesthetic (different patterns inside and outside) or function (insertable/removable filters). The authors would pose that data demonstrating combinations of materials providing increased protection for wearers would likely find a sizeable audience who would accept the increased complexity of construction and decrease in Δp for better filtration performance. A sentence addressing that most surgical masks and respirators are already multi-layer constructions was added to the discussion section.

(6) In the Conclusions part, the authors said that “When rapid in-house testing of material filtration is needed, hospitals with quantitative fit-testing equipment can create a setup that approximates NIOSH testing standards using <$300 of additional equipment”. Whether the test result is accurate? The paper should provide some evidence. Some analysis should be added in the Results part.

As discussed above, the authors agree that the validation and correlation of measurements from the presented test setup with those from NIOSH can be emphasized beyond what is already in the discussion section. Additional language has been provided in the results section. The conclusion has been modified to reflect these additions. 

Reviewer #2: The manuscript focused on the measurement of filtration efficiencies using modified respirator fit tester setup. The research is technically sound and properly organized. The experiments are conducted rigorously and the conclusions are drawn appropriately. Add some keywords if necessary.

We appreciate the second reviewer’s time and attention to our manuscript. We appreciate his or her comments.

---

## [Decision Letter · Decision Letter 1]

29 Sep 2020

Measurement of filtration efficiencies of healthcare and consumer materials using modified respirator fit tester setup

PONE-D-20-22953R1

Dear Dr. Long,

We’re pleased to inform you that your manuscript has been judged scientifically suitable for publication and will be formally accepted for publication once it meets all outstanding technical requirements.

Kind regards,

Ming Zhang, Ph.D.

Academic Editor

PLOS ONE

Additional Editor Comments (optional):

Reviewers' comments:

Reviewer's Responses to Questions

**Comments to the Author**

1. If the authors have adequately addressed your comments raised in a previous round of review and you feel that this manuscript is now acceptable for publication, you may indicate that here to bypass the “Comments to the Author” section, enter your conflict of interest statement in the “Confidential to Editor” section, and submit your "Accept" recommendation.

Reviewer #1: All comments have been addressed

2. Is the manuscript technically sound, and do the data support the conclusions?

Reviewer #1: Yes

3. Has the statistical analysis been performed appropriately and rigorously? 

Reviewer #1: Yes

4. Have the authors made all data underlying the findings in their manuscript fully available?

Reviewer #1: Yes

5. Is the manuscript presented in an intelligible fashion and written in standard English?

Reviewer #1: Yes

6. Review Comments to the Author

Reviewer #1: (No Response)

7. PLOS authors have the option to publish the peer review history of their article (what does this mean?). If published, this will include your full peer review and any attached files.

Reviewer #1: No

---

## [Editor Report · Acceptance letter]

5 Oct 2020

PONE-D-20-22953R1 

Measurement of filtration efficiencies of healthcare and consumer materials using modified respirator fit tester setup 

Dear Dr. Long:

I'm pleased to inform you that your manuscript has been deemed suitable for publication in PLOS ONE. Congratulations! Your manuscript is now with our production department. 

Kind regards, 

on behalf of

Dr. Ming Zhang 

Academic Editor

PLOS ONE